# Lessons learned from the adaptation of the Reflective Functioning Questionnaire (RFQ) for Dutch people with mild to borderline intellectual disabilities

Suzanne D. M. Derks[1]*, Agnes M. Willemen[1], Cis Vrijmoeth[2], Paula S. Sterkenburg[1,3]

1 Department of Clinical Child and Family Studies & Amsterdam Public Health, Vrije Universiteit Amsterdam, Amsterdam, The Netherlands, 2 Centre for Research and Innovation in Christian Mental Health Care, Hoevelaken, The Netherlands, 3 Bartiméus, Doorn, The Netherlands

☯ These authors contributed equally to this work.

* s.d.m.derks@vu.nl

## Abstract

### Background

People with mild to borderline intellectual disabilities (MBIDs) face challenges in social functioning, possibly as a result of limited mentalising abilities such as reflecting on the behaviour of themselves and others. Reflective functioning in people with MBIDs has not yet been investigated due to a lack of instruments. The Reflective Functioning Questionnaire (RFQ) is a seemingly easy adaptable, short self-report questionnaire. The aim of the present, explorative study was to adapt the RFQ for people with MBIDs and investigate the psychometric properties and correlations with other mentalising related constructs. The formulation of the items was adapted to the target group and items were added to broaden the scope towards reflection on both the self and other.

### Method

Participants were 159 adults with MBIDs who completed a Dutch-translated and easy-to-read RFQ with five supplemental items, a questionnaire for autistic traits, a self-report questionnaire assessing perspective taking and two performance-based measures assessing emotion recognition and Theory of Mind.

### Results

Confirmatory factor analysis confirmed the factor structure of the RFQ and revealed a two-factor structure with a Self and Other subscale. Generally satisfactory internal consistency and test-retest reliability were found. Explorative results showed correlations of the RFQ-8 and RFQ subscales with autistic traits and between the RFQ Other and perspective taking.

**Data Availability Statement:** Data cannot be shared publicly because of legal restrictions; that is, the data contain sensitive information of a

scarce, vulnerable population. Requests for access to our data can be made to the data and management secretary of the Academic Lab Bartiméus - Vrije Universiteit Amsterdam, Joyce Schroor, +31 6 18 18 90 59, j.schroor@vu.nl.

**Funding:** This work was funded by The Netherlands Organization for Health Research and Development ZonMw, Postbus 93 245, 2509 AE Den Haag The Netherlands. Project number 845004004. The funders had no role in study design, data collection and analysis, decision to publish, or preparation of the manuscript.

**Competing interests:** The authors have declared that no competing interests exist.

## Conclusions

This explorative study is the first testing psychometric properties of the RFQ as a self-report questionnaire for assessing reflective functioning in adults with MBIDs. This step is relevant in gaining more scientific knowledge on assessing mentalising in people with MBIDs.

## Introduction

Since Fonagy [1] reintroduced the concept of mentalisation in the 1990s, a growing body of literature has covered the development, use, and improvement of mentalising. Research shows that children acquire mentalising in secure attachment relationships with parents and caregivers during early childhood [2], and that this is necessary for building fundamental social relationships and interactions [3]. For people with mild to borderline intellectual disabilities (MBIDs), mentalising may be atypical because of reduced cognitive abilities [4], impeding social relationships. To gain insight into the proper measurement of mentalising, a instrument for measuring this in people with MBIDs is required. However, such an instrument for this target group has not yet been developed or assessed.

Mentalisation can be defined as a person's ability to see, understand, and reflect on their own behaviour and that of others in terms of intentional mental states with intentional meanings, such as desires, needs, wishes, and feelings [2, 5]. This is a quintessential capacity for acquiring a stable sense of self and reciprocal social relationships that develops over time in secure caregiver–child attachment relationships, with parents or caregivers attuning their reactions emotionally to the state of the infant and facilitating the child's learning about feelings [3]. In its content and process, the ability to mentalise spans four dimensions or components, namely: the cognitive and affective aspect; the self and others as objects; the automatic or controlled modes of functioning; and the internal and external attention distinction [6]. Different neural circuits are involved in using these components and switching between them [7]. Several concepts are associated with mentalising, such as reflective functioning (internal and self-oriented), perspective taking (internal and other-oriented) and Theory of Mind and emotion recognition (external and other-oriented) [2].

Mentalisation is not a fixed ability, but a dynamic, influential skill, and its development is related to the development in other domains, such as language and cognition [2]. Therefore, atypical development in these domains may be related to impaired mentalising skills, such as in the ability to read one's own mind and that of others [7]. Furthermore, the presence of autistic traits may impede development and suggest limitations on mentalising skills or their manifestation, especially in mentalising related to others' thoughts and feelings [8].

Specific limitations in several social-cognitive skills have been reported for people with MBIDs, such as reflection, perspective taking, contextualising self in relation to others' thoughts and feelings, recognising and identifying feelings and emotions, and accurately assessing facial expressions, emotions, and the intentions of others [4, 9, 10]. People with deficits in general intellectual functioning (IQ 50–85) and adaptive functioning compared to typically developed peers can be classified as having MBIDs, including both borderline intellectual functioning (BIF) and mild intellectual disabilities (MIDs) [11]. Both groups experience similar challenges in mentalising and social functioning [10, 12]. It has been emphasised that people with BIF should also be included in care, treatment, and research in order to avoid possible risks of under-recognition, misclassification, and falling between the cracks [13]. In the Netherlands, people with BIF are intentionally included in health care for and research of people

with intellectual disabilities [14]. Therefore, in this study, when we refer to people with mild intellectual disabilities (MBIDs), both MID and BIF are intended.

Instruments are needed to gain more insight into mentalising in people with intellectual disabilities and provide proper support for any necessary development. Fonagy et al. [15] developed an interview-based measure to assess an important cognitive process underlying the capacity to mentalise, namely reflective functioning. Although this measure is examined on psychometric properties, its use in an empirical context is limited by practicalities, as it is time-consuming and costly to use (e.g., highly trained scorers are required [16]). To overcome these limitations, Fonagy et al. [17] developed a self-report measure of reflective functioning for people with borderline personality disorder (BPD), the Reflective Functioning Questionnaire (RFQ). Fonagy et al. [17] developed a 46-item and 54-item version before settling on a final 8-item version (RFQ-8). The RFQ consist of 8 items divided across two subscales: 'Certainty about mental states' and 'Uncertainty about mental states'. The Certainty subscale assesses the development of complex models of the mind that are inconsistent with observable evidence (i.e., hypermentalising). The Uncertainty subscale assesses the great difficulty with developing complex models of the mind of the self and/or others (i.e., hypomentalising). Genuine mentalising is the optimal level of mentalising in between hyper- and hypomentalising, characterised by a balanced stance of knowing and not always knowing the mental states of themselves and others [17]. Four of the eight items were used in both subscales. Therefore, these items were double scored in opposite direction. However, this double scoring resulting in a two-factor structure is psychometrically questionable because items in a factor analysis are assumed to be independent. These challenges were noted in the studies of Müller et al. [18], Spitzer et al. [19] and Woźniak-Prus et al. [20]. Therefore, the authors took a step back and conducted an exploratory factor analysis to assess factor structure of the RFQ. The authors showed that the eight items fit well as a unidimensional construct [18–20].

Given that the RFQ is a short self-report instrument with items that seemed to adapt well to other groups, the question was raised as to whether the RFQ can also be used in people with MBIDs. Although other instruments exist, such as The Mentalization Scale (MentS) [21], the RFQ-8 is relatively short and appear psychometrically sound (e.g., [18–20]). Accordingly, the first aim of the present study was to explore the psychometric properties of the RFQ-8 adapted for people with MBIDs. As also concluded by Müller et al. [18], the RFQ-8 is mainly focused on consideration of the self and one's own feelings and thoughts. However, as mentioned above, people with MBIDs face challenges in reflecting on both the self and others. Therefore, including items with a broader focus on the self in relation to the feelings and thoughts of others, which are available in the RFQ-54, may support the suitability of the RFQ for people with MBIDs. Thus, the second aim of this study was to explore the psychometric properties of an extended RFQ.

First, we investigated the proposed one-factor structure of the RFQ-8 as suggested by Müller et al. [18] and Woźniak-Prus et al. [20]. Second, in co-creation, we selected some additional items from the 54-item version of the RFQ that focus on the self in relation to the thoughts and feelings of others. We investigated the factor-structure including the additional items. Next, we evaluated the internal consistency and test-retest reliability of both versions. Third, correlations are examined, comparing the RFQ-8 and the extended RFQ with autistic traits. Stronger correlations were expected for the extended RFQ and autistic traits, because the other-oriented focus is also represented in autistic traits (e.g., greater challenge with other people's thoughts and feelings). In addition, associations between the RFQ-8 and the extended RFQ and perspective taking, emotion recognition and Theory of Mind were investigated. Weaker correlations with the RFQ-8 compared to the extended RFQ are expected as the RFQ-8 is primarily internal and self-oriented while the other concepts reflect the external, other-

oriented (e.g., emotion recognition and Theory of Mind) or internal, other-oriented (e.g., perspective taking) dimensions of mentalising.

## Materials and methods

### Design and participants

A total of 159 adults with MBIDs participated in this study. Inclusion criteria were age $\geq$ 18 years and a known indication of MBIDs. Persons with a visual or auditive impairment were included, but persons who were deaf and/or blind were excluded. Participants were not asked to report their exact IQ test scores, as participants had to be able to participate independently and frequently do not know their IQ scores by themselves. Table 1 shows the demographic characteristics of the sample.

Post-hoc power analysis revealed that a total sample size of 159 participants yielded sufficient statistical power ($\beta$ = .80) to detect a small to medium effect size of $r \geq$ .20 (G*power 3.1.9.4) [22]. According to the ratio of 10 participants per item [23], our sample size was sufficient to conduct a factor analysis on the RFQ-8 and for the analysis with the additional items (up to 5 additional items). Test-retest sample size ($n$ = 83) was satisfactory to detect a minimum acceptable ICC of .50, with an expect ICC of .70, $\alpha$ = .05 two-tailed, $\beta$ = .80, $k$ = 2 and expected dropout rate of 10% [24].

### Procedure

Participants were recruited from care organisations in the Netherlands (i.e., ASVZ, Bartiméus, Cordaan, Ons Tweede Thuis). Independent research assistants supported the participants in completing the digital questionnaire following a standardised protocol. According to this protocol, research assistants were instructed to help the participants with the digital

**Table 1. Demographic variables of the participants ($N$ = 159).**

| | | |
|---|---|---|
| Gender | Male | 55% |
| | Female | 45% |
| Age, years | 19–29 | 43% |
| | 30–39 | 21% |
| | 40–49 | 18% |
| | 50–59 | 12% |
| | 60–69 | 6% |
| Education | None | 6% |
| | Special primary education | 26% |
| | Primary education | 4% |
| | Special secondary education | 36% |
| | Secondary education | 16% |
| | Vocational secondary education | 9% |
| | Other | 3% |
| Receiving care | 24/7 | 54% |
| | Less than 24/7 | 46% |
| Work and/or day care | Work (paid to voluntary) | 45% |
| | No work (school to home stay) | 55% |
| Legal representative | Mentor | 72% |
| | Administrator | 80% |
| | Curator | 20% |

questionnaires provided by Qualtrics software (i.e., digital support) and to stimulate thinking through extra explanations (see S1 File). Moreover, they appointed that it is about their thoughts and feelings and that there were no wrong answers. In addition, participants are ensured that their answers would not be shared with other people. If escalating situations arose, a professional caregiver was available but did not interfere with the procedure. A total of 159 participants filled out the questionnaires. Five weeks later, a randomly selected subsample ($n$ = 83) was asked to fill out the RFQ a second time for test-retest analyses. No significant differences were found between the sample and the subsample regarding demographic variables ($p$ > .05).

All participants provided written informed consent prior to the assessment. In cases of legal incapacitation, their legal representative provided consent. The consent form was adapted to the level of comprehension of the participants. In addition, the independent researchers supported the participants if necessary. Independent researchers signed a confidentiality agreement. Ethical approval was provided by the Medical Ethics Committee of the University Medical Centre Amsterdam location VUmc, the Netherlands (METc VUmc 2018.007, NL.60353.029.17) and the Institutional Review Board of the Faculty of Behavioural and Movement Sciences of the Vrije Universiteit Amsterdam (VCWE-2017-171).

## Instruments

**The Reflective Functioning Questionnaire (RFQ).** The original (8-item) self-report version of the RFQ [17] was used to measure internal mentalising through reflective functioning. Items were scored on a 7-point Likert scale ranging from *strongly disagree* (1) to *strongly agree* (7). As advised by Finlay and Lyons to break a question into two stages for people with MIBDs [25], the answering options on the 7-point scale were split into two steps. First, participants could choose to score disagree, neutral, or agree. Second, the choices 'disagree' and 'agree' were split into strongly, quite a bit, and somewhat.

To increase the relevance of the RFQ to the target group, items were added by a two-step process. First, two members of the project team independently studied the 54-item version of the RFQ using Choi-Kain and Gunderson's dimensions of mentalisation [6], cognition, affection, the self, and the other, and then highlighted the items with a focus on the feelings and thoughts of others, resulting in 14 eligible items. Second, to keep the number of supplemental items as low as possible, the 14 items were thoroughly screened for overlap and comprehensibility and applicability to the target group. For example, the items 'It's easy for me to figure out what someone else is thinking or feeling' and 'It's really hard for me to figure out what goes on in other people's heads' were both highlighted. However, as the first item is more concrete, that is desirable for people with intellectual disabilities [25], this item was chosen to include. Eventually, this resulted in five items (original RFQ-54 items 2, 22, 26, 42 and 43) that were potentially valuable to add. In this study, we investigated whether adding five items to the RFQ is of value.

All 16 items were translated and adapted following guidelines from the World Health Organisation [26]. The steps were as follows: 1) two members of the project team and a scientific practitioner familiar with people with MBIDs performed an individual forward translation (English to Dutch) and comparison of the translations; 2) items were further adapted by means of simplifying concepts and removing unnecessary wording until consensus was reached; 3) an independent researcher back-translated the items (English to Dutch); 4) adaptations were checked for acceptability against the original instrument; and 5) co-researchers, three adults with MBIDs, provided first feasibility and final pre-testing of questionnaires and final feedback on the items. In addition, the back-translated items are checked on

comprehensibility by a native English professional and client in the care for people with intellectual disabilities (see S1 File for the questionnaire).

Both the 8 items of the original RFQ and the 5 additional items are coded as the original scoring of 1,2,3,4,5,6,7, in which a higher score represented more uncertainty on mentalising. Therefore, original item 7 and original RFQ-54 items 2, 42 and 43, included as additional items 9, 12 and 13, are recoded because of reversed scoring. Recent studies have indicated adequate psychometric properties of the original RFQ-8 as unidimensional scale in a German inpatients sample, a German young adult sample, a United States adult sample [18] and a German adult sample [19], with McDonald's omegas of .79, .82, .87 and .82 respectively. Additionally, a Cronbach's alpha of .75 was found in a Polish sample of students [20].

**Autistic traits.** The Autism Spectrum Quotient (AQ-10) [27] consists of 10 items measured on a 4-point Likert scale, with scores ranging from *definitely agree* (1) to *definitely disagree* (4). To adapt the Dutch version [28] of the questionnaire for adults with MBIDs, we simplified the concepts, and the co-researchers with MBIDs checked them. Items were for example 'I find it easy to see from someone's face what someone thinks or feels' and 'I find it hard to understand what others mean'. A mean score was calculated for the 10 items, with higher scores indicating more autistic traits. In their study, Allison et al. [27] showed that the AQ-10 has good psychometric properties in adults, with a Cronbach's alpha of .85. In the current study, Cronbach's alpha was .42. Further analyses did not show such an improvement in scale when omitting individual items, which has led to sticking with the original scale.

**Perspective taking.** The Subscale Perspective-Taking (PT) of the Interpersonal Reactivity Index (IRI) [29] was used to measure the internal mentalising skill of perspective taking, the ability to take the psychological point of view of others. The subscale consisted of seven items rated on a 5-point Likert scale ranging from *does not describe me well* (1) to *describes me very well* (5). The Dutch translation of the subscale was used [30] and adapted for adults with intellectual disabilities by removing unnecessary wording and simplifying concepts. Another researcher and co-researchers checked the adaptations, resulting in minor changes, such as dividing one sentence into two. The text was also back translated by a native English-speaking researcher on the team who did not know the instrument. The English back-translation was then checked by a native English-speaking person who is an expert in English 'easy to read language'.

With a Cronbach's α of .71 for the complete IRI tested in a sample of adults with MBIDs [31], the subscale was expected to be adequately reliable for this population. The current study found a Cronbach's alpha of .54 among adults with MBIDs. After closer inspection of the items, the two items 'I sometimes find it difficult to see things from the "other guy's" point of view' and 'If I'm sure I'm right about something, I don't waste much time listening to other people's arguments' seemed to be problematic. These two items were negatively phrased, which is more challenging for people with intellectual disabilities [25]. After excluding these items, Cronbach's alpha increased to a fair level of .64 [32]. Therefore, the mean score for perspective taking was based on the five remaining items, with higher scores reflecting higher levels of perspective taking.

**Emotion recognition.** The Radboud Faces Database (RaFD) [33] assessed emotion recognition as part of external, other-oriented mentalisation. Participants had to view colour photographs of unfamiliar faces of adults portraying different emotional expressions with different gaze directions. For each photograph, participants had to choose one of five options that best fit the intended emotion. A mean score for each participant was calculated based on the number of incorrect (0) and correct (1) answers, with higher scores indicating more correct answers being given. To use the RaFD for people with intellectual disabilities, 50 photographs were selected based on average genuineness, intensity, and clarity of the emotion, mean

valence of the photographs, and percentage agreement on emotion categorisation present in the assessment study of the RaFD [34]. With an agreement rate of 82% between chosen and intended expressions (median 86%, $SD$ = 12.8%, $N$ = 149), the RaFD had good psychometric qualities in the current group of adult participants with MBIDs.

**Theory of mind.** The Frith–Happé animations test [35] assesses attributions of external, other-oriented mental states through a nonverbal task. Participants watch a series of silent computer animations. Each animation lasts 34–45 seconds and, after each animation, participants are asked, "What was happening in the animation?" Answers to this question (verbal descriptions) were audiotaped while completing this test and scored for the level of complexity of mental state terms used (intentionality of the answer, scoring 0–3). Mean intentionality scores were calculated, with higher average scores reflecting more complexity of the mental state terms used. The test has been used with people with intellectual disabilities and has adequate face-validity with no identified ceiling effects [35].

In the current study, two independent coders who were not involved in administering the questionnaires received training in using the scoring criteria. They then scored all recorded verbal descriptions of participants. Intraclass Correlation Coefficients (ICCs) between the scores were calculated. When the coders differed by more than 1 point, they discussed the result ($n$ = 24), and a joint score was reached. The final inter-rater reliability was adequate (ICC = .804).

## Data analysis

Data analyses were performed using SPSS (version 26). Demographic variables (intellectual disability, sex, age, work and/or day care) were explored. Missing values were analysed using Little's MCAR test, showing that missing values were completely at random ($\chi^2$ (21, $N$ = 158) = 31.6, $p$ = .064). Missing data were handled using pairwise deletion.

Factor structure was determined first by conducting a confirmatory factor analysis (CFA) in AMOS (version 26) for the unidimensional model with the RFQ-8 adapted for people with MBIDs. The goodness of fit of factor structures was determined with the normed chi-square (< 2 considered good, ≥ 2 to < 3 considered acceptable), root mean square error of approximation (RMSEA; < .05 considered good, values between .05 and .08 considered acceptable), comparative fit index (CFI; values between .90 and .95 considered acceptable, values > .95 considered good), and standardised root mean square residual (SRMR; values < .10 considered good) [36]. Second, exploratory factor analysis (EFA) was conducted to examine the factor structure of the RFQ including the additional items. EFA was performed with Principal Components analysis. Scree plot and parallel analysis [37] were used to determine the number of factors. When applicable, the factor structure found in the EFA was evaluated in a CFA.

To assess internal consistency, we calculated the Cronbach's alphas and McDonald's omegas for the RFQ-8 and a potential new RFQ with additional items (alphas and omegas could be either unsatisfactory, fair, moderate, good or excellent following the guidelines of Ponterotto & Ruckdeschel [32]). In addition, we assessed the test-retest reliability using ICCs [mean rating ($k$ = 2), consistency agreement, 2-way mixed-effects model, 95% confidence interval (CI)] by using test-retest data. Values can indicate poor (< .50), moderate (≥ .50 to < .75), good (≥ .75 to < .90), and excellent (≥ .90) reliability [38].

In addition, Pearson correlations were first computed of the RFQ-8 and the extended RFQ (including items related to the self and the other) with the AQ-10 and second of the RFQ-8 and the extended RFQ with the PT subscale of the IRI, the RaFD, and the Frith–Happé animations test. In addition, correlations were compared using the method developed by Steiger [39] and updated by Hoerger [40].

## Results

### CFA of factor structure

First, the CFA showed a good model fit for the unidimensional RFQ-8 [$\chi^2$/df = 1.39; RMSEA = .05 (95% CI = .00–.09); CFI = .96, SRMR = .05]. As the results in Fig 1a shows, items 1 and 7 had the lowest standardised factor loadings.

Second, EFA was performed with the RFQ-8 together with the five supplemental items. First, scree plot analyses en parallel analysis revealed a two-factor structure. The Kaiser-Meyer-Olkin Measure of Sampling Adequacy (KMO = 0.707) and Bartlett's Test of Sphericity ($\chi^2$(78) = 408.2; $p < .001$) confirmed appropriateness of the EFA. The results of the EFA are presented in Table 2. Content analysis of the items moreover indicated that, with the two-factor structure, an interesting distinction seemed to be made between 1) items that represent the self and one's own feelings and thoughts and 2) items that represent the self in relation to the feelings and thoughts of others. In accordance with the results in Müller et al. [18] and Woźniak-Prus et al. [20] and the above result, item 7 exhibited as an outsider with a weak factor loading. Its fit was checked in the CFA. In addition, item 13 showed extremely weak factor loadings on both factors. Therefore, item 13 was yet omitted.

Next, the factor structure found in the EFA was fitted in a CFA. This two-factor structure showed poor model fit [$\chi^2$/df = 2.50; RMSEA = .10 (95% CI = .08–.12); CFI = .77, SRMR = .09]. Items 7 and 12 showed factor loadings lower than .03. Omitting items 7 and 12, model fit improved to acceptable fit [$\chi^2$/df = 1.78; RMSEA = .07 (95% CI = .04–.10); CFI = .91, SRMR = .07; Fig 1b]. In accordance with the factors found in the EFA, the CFA confirmed a two-factor structure with a factor 'RFQ Self', representing reflection on the self and one's own feelings and thoughts and a factor 'RFQ Other', representing reflection on the self in relation to the feelings and thoughts of others. These factors resulted in two subscales by averaging the six items that loaded on the Self factor and averaging the four items that loaded on the Other factor.

### Internal consistency and test-retest reliability

The internal consistency of the unidimensional RFQ-8 was found to be fair (N = 158), with a Cronbach's alpha of .71 and a McDonald's omega of .73 (8 items). Individual item analysis showed that with deleting item 1 and 7, Cronbach's alpha and McDonald's omega increase to .74 and .75 respectively, forming the identical RFQ Self (6 items) with moderate internal consistency. The internal consistency of the RFQ Other (4 items) was found to be unsatisfactory with an alpha of .60 and omega of .62.

Five-week interval test-retest reliability was found to be moderate (n = 83), with an ICC of .722 (95% CI = .571–.820) for the unidimensional RFQ-8. The test-retest reliability of the RFQ Self and RFQ Other was moderate, with ICCs of .699 (95% CI = .534–.805) and .637 (95% CI = .438–.765) respectively.

### Correlations

Table 3 provides an overview of mean scores, standard deviations, and ranges for the RFQ and other measures of mentalisation and the Pearson correlations between all variables. The unidimensional RFQ-8 and the RFQ Self were positively and significantly related to the AQ-10, but not significantly correlated to the PT subscale, the RaFD and the Frith–Happé animations test. The RFQ Other was also positively and significantly correlated to the AQ-10, but in contrary to the unidimensional RFQ-8 and the RFQ Self, significantly and negatively correlated to the PT subscale. In addition, the RFQ Other was not significantly correlated to the RaFD and the

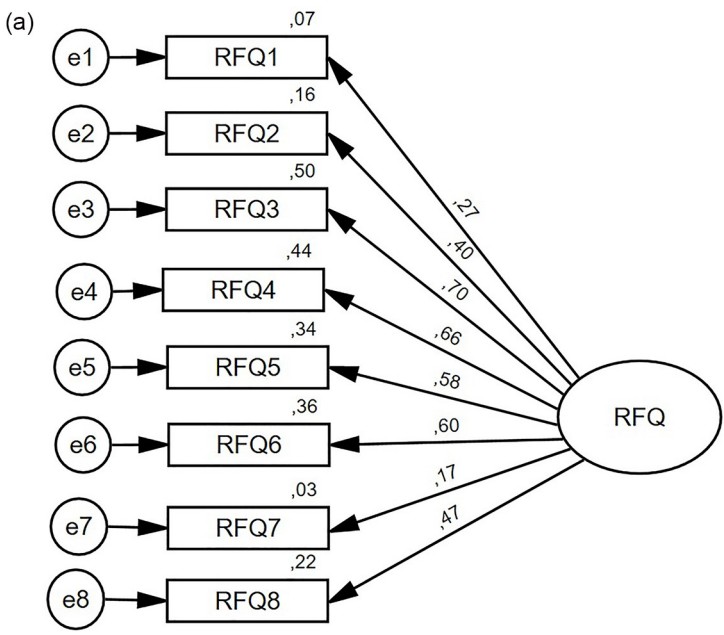

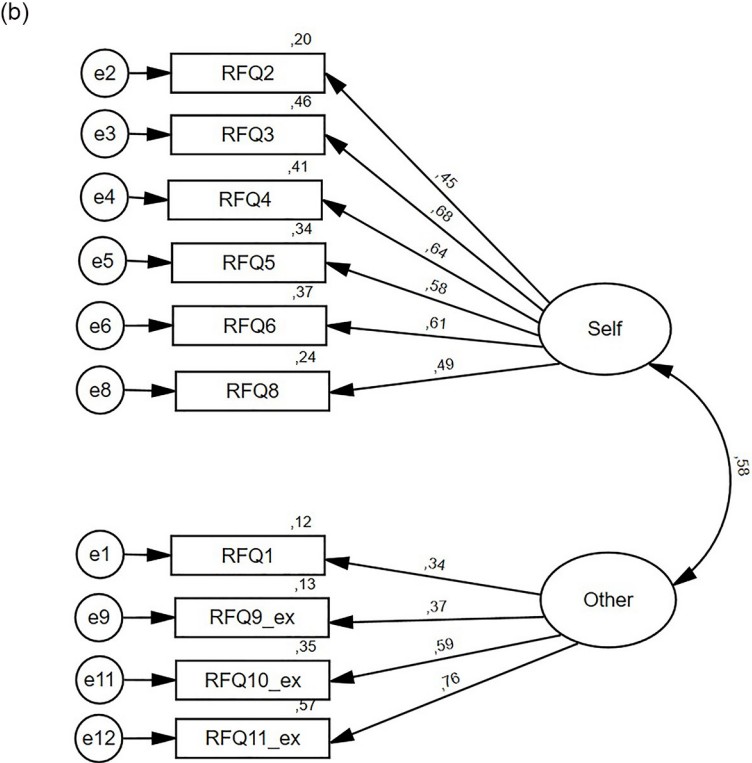

**Fig 1.** a and b. Visual representation of the unidimensional factor structure of the RFQ-8 and the two-factor structure of the Reflective Functioning Questionnaire including three additional items in a sample of 159 adults with mild to borderline intellectual disabilities. Fig 1. Visual representation of the unidimensional and new two-factor structure of the Reflective Functioning Questionnaire for 159 adults with mild to borderline intellectual disabilities. The ovals indicate the factors, and the rectangles indicate the items. Residuals (expressed as covariance) are indicated by the numbers to the left of the rectangles. Hypothesised direct effects (expressed as standardised regression coefficients) are indicated by the numbers between the single-arrow lines. A relationship between the factors (expressed as correlations) is implied by the number on the right between the bidirectional arrows.

**Table 2. Factor loadings for the RFQ-8 and additional items (N = 159).**

| Item | | Factor 1 | Factor 2 |
|---|---|---|---|
| RFQ1 | I don't understand what people think. | 0.269 | **0.353** |
| RFQ2 | I don't always know why I do something. | **0.499** | 0.294 |
| RFQ3 | When I am angry, I say things without knowing why. | **0.697** | 0.081 |
| RFQ4 | When I am angry, I say things I don't want to say. | **0.722** | -0.092 |
| RFQ5 | When I don't feel safe, I sometimes do things that other people don't like. | **0.668** | -0.047 |
| RFQ6 | Sometimes I do things and I don't know why. | **0.693** | -0.020 |
| RFQ7 | I always know what I feel. | 0.156 | **0.325** |
| RFQ8 | Strong feelings make it hard to think. | **0.573** | 0.087 |
| RFQ9_ex | It is easy for me to know what other people are thinking and feeling. | -0.041 | **0.826** |
| RFQ10_ex | I find it hard to understand other people's feelings. | 0.334 | **0.486** |
| RFQ11_ex | The thoughts and feelings of other people confuse me. | 0.527 | **0.504** |
| RFQ12_ex | I often know what other people think. | -0.220 | **0.720** |
| RFQ13_ex | I know my feelings can change. Even if it doesn't feel that way. | -0.152 | 0.259 |

*Note*. Factor loadings higher than 0.3 are bold, indicating factor structure.

Frith–Happé animations test. Comparing the correlations of the RFQ subscales, the RFQ Other correlated significantly stronger with the AQ-10 (Steiger Z = 3.07, p = .002) and the PT subscale (Steiger Z = 2.80, p = .005) than the RFQ Self.

## Discussion

The results of the current study showed adequate model fit for the unidimensional RFQ-8, as proposed by Müller et al. [18] and Woźniak-Prus et al. [20]. In addition, an adequate model fit was found for a two-factor structure with the 'RFQ Self', representing reflection on the self and one's own feelings and thoughts, and 'RFQ Other', representing reflections on the self in relation to the feelings and thoughts of others. The internal consistency was fair for the unidimensional original RFQ-8, moderate for the RFQ Self and unsatisfactory for the RFQ Other. Test-retest reliability was moderate for the original scale and also for the formed subscales. Stronger correlations between the RFQ Other and respectively autistic traits and perspective taking compared to the unidimensional RFQ-8 or RFQ Self were found.

The two-factor structure with the 'RFQ Self' and 'RFQ Other' is in line with recent findings of Müller et al. [18], who developed a new mentalising measure called the Certainty About

**Table 3. Means, standard deviations, and correlations among the mentalisation measurements (N = 159).**

| Measure | Mean | SD | 1 | 2 | 3 | 4 | 5 | 6 | 7 |
|---|---|---|---|---|---|---|---|---|---|
| 1. Unidimensional RFQ-8 | 4.02 | 1.14 | | | | | | | |
| 2. RFQ_Self | 4.26 | 1.35 | .968** | | | | | | |
| 3. RFQ_Other | 3.97 | 1.28 | .465** | .369** | | | | | |
| 4. Autistic traits | 2.49 | 0.40 | .249** | .185* | .441** | | | | |
| 5. Perspective taking | 3.67 | 0.83 | -.032 | .032 | -.217** | -.234** | | | |
| 6. Emotion recognition | 0.82 | 0.13 | -.015 | .014 | -.114 | .054 | .145 | . | |
| 7. Theory of Mind | 1.17 | 0.40 | .094 | .117 | -.026 | .053 | .073 | .361** | |

* p < .05

** p < .001

Mental States Questionnaire (CAMSQ) [41]. Both results underline the importance of including the dimensions of 'the self' and 'the other' when assessing mentalising. Creating a specific RFQ Other for people with MBIDs seems moreover valuable as they tend to have more difficulties reflecting on the feelings and thoughts of others and taking the perspective of others [42, 43].

Fair internal consistency for the unidimensional RFQ-8 is consistent with the finding reported by Woźniak-Prus et al. [20]. Individual item analysis furthermore pointed in the direction of creating the RFQ Self with the suggestion to remove items 1 and 7, that moreover is consistent with indications of Müller et al. [18] and Woźniak-Prus et al. [20]. In addition, internal consistency did not improve in the RFQ Self or RFQ Other. A general explanation for relatively low internal consistencies of the RFQ might be that the operationalisation of the underlying construct is somewhat problematic. Higher scores reflect more uncertainty in reflective functioning, which is equivalent to a difficulty in developing complex models of the mind of oneself and/or others. However, to measure this in the RFQ, people answer questions that still require a degree of reflection and assessment, especially problematic for people with MBIDs due to limitations in intellectual functioning [43]. This indicates a contradiction that is most evident in the internal consistency and may restrict the interpretation of the scores on the scales in situations where self-reflection may be limited.

Five-week interval test-retest ICCs were all moderate. The unidimensional RFQ-8, the RFQ Self and the RFQ Other identified two items that seem prone to reconsideration: "*I don't always know why I do something*" (in the unidimensional RFQ-8 and the RFQ Self) and "*I find it hard to understand other people's feelings*" (in the RFQ Other). People with MBIDs can have more difficulties with negatively formulated questions in combination with negatively formulated answer categories, resulting in a double denial [25], which can lead to variation or error in the test-retest measurements. Furthermore, reconsiderations are more likely when addressing more extreme situations [44]. This explanation appeared to apply more specifically to items 2 and 10 (i.e., using the words "*always*" and "*hard*"), which is more likely to be reconsidered and scored differently when asked a second time.

Correlations between autistic traits and the RFQ-8 were in line with correlations found among adult patients with Borderline Personality Disorder (BPD) [45]. As expected, we found positive, significant correlations with all the scales, especially for the RFQ Other. The findings imply that the presence of autistic traits was associated with more uncertainty about thoughts and feelings of the self and even more of others. This is in accordance with research findings that shows that people with autistic traits have less genuine reflections on the self and others [2], which may result in experiencing difficulties in social interaction and communication.

Furthermore, non-significant correlations between the unidimensional RFQ-8 and the RFQ Self and perspective taking, emotion recognition and Theory of Mind were found. This can indicate that the primarily internal, self-oriented focus of reflective functioning in the unidimensional RFQ-8 and the RFQ Self on the one hand and the external and internal other-oriented focus of perspective taking, emotion recognition and Theory of Mind on the other hand refer to different domains of mentalising [2, 5, 6]. With the addition of the other-oriented items in the RFQ Other, both the RFQ Other and the perspective taking measure had an internal, other-oriented focus, explaining the correlation found. In sum, this study shows that reflective functioning measured with the RFQ can provide complementary insights about mentalising of a certain person when added to a battery of other measures assessing important skills for mentalising. However, following Müller et al. [18], Spitzer et al. [19] and Woźniak-Prus et al. [20], the RFQ alone does not seem to be sufficient to assess (the lack of) complex mentalising, also for people with MBIDs.

## Strengths and limitations

To the best of our knowledge, this study represents the first attempt to investigate psychometric properties of the RFQ for assessing reflective functioning about mental states in adults with MBIDs. With two created subscales, the RFQ Self and RFQ Other, this study encourages to replicate the findings in other samples. The findings are promising but also need to be interpreted in the context of some limitations. First, considering a small effect size (.10), the sample size of the study ($N$ = 159) is not sufficient to establish significant correlations (to .40) [46]. Additional data of adults with MBIDs is necessary to strengthen the correlations. Second, given the novelty of the study, the correlational analyses are restricted by unmodified (AQ-10, PT subscale) and unvalidated (AQ-10, PT subscale, RaFD, Frith Happé animations test) instruments for people with MBIDs. As these instruments were assessed for the first time in this target group, results also could and did reveal challenges with the instruments (i.e., low internal consistency). This has limited statements about the psychometric properties of the RFQ. In addition, because measuring elements of complex mentalising in a patient-reported outcome measure such as the RFQ also carries a high risk of overestimation [44], performance-based outcome measures can be a valuable addition as a counterbalance to patient-reported measures. Studies should further investigate psychometric properties of (other) measures of mentalising. For example, measures that still need to be adapted and validated to adults with MBIDs, such as the experimental Movie for the Assessment of Social Cognition (MASC) [47]. Third, it was beyond the scope of this study to examine other associations between the RFQ and instruments measuring (developmental) concepts, such as social-emotional development [48], epistemic trust [49], or personal distress [50]. Moreover, as recently discussed by Müller et al. [18], some items of the RFQ may overlap with characteristics such as impulsivity and emotional lability, questioning the accuracy of the RFQ to measure mentalising. With this in mind, Müller et al. [18] developed the CAMSQ [41]. Future studies are invited to first replicate this study and investigate the convergence of the RFQ with impulsivity, and second to examine the applicability of new questionnaires as the CAMSQ [41] for people with MBIDs. Fourth, although all participants had an indication of MBIDs (IQ 50–85) confirmed before the start of the study, the population of adults with MBIDs is very diverse. However, the level of comprehension was not specifically examined in our study. Future studies may need to examine associations between IQ and the level of comprehension tested with for example Wechsler Adult Intelligence Scale (WAIS) [51] and the usability of the RFQ. Fifth, the selection of the additional, other-oriented items was consciously made to add the least but most valuable items as possible for adults with MBIDs. However, with two items left out that did not fit in the scale, the created RFQ Other has only four items. To extend the RFQ Other, more other-orientated items from the RFQ-54 can be added in future studies. Finally, with a low Cronbach's alpha for the AQ-10, the chance of finding non-significant effects increased in the analyses (Type II errors) [32]. In addition, Müller et al. [18], Spitzer et al. [19] and Woźniak-Prus et al. [20] also examined U-shaped associations between the RFQ and maladaptive characteristics such as anxiety, signs of psychopathology and mental health problems, since very low scores on the RFQ may indicate hypermentalising (e.g., being overconfident in understanding mental states). Future studies are invited to perform analyses to further investigate the possibility of a U-shaped association for people with intellectual disabilities with a well-validated measure of autistic traits.

## Implications for practice

These findings contribute to the assessment of mentalising in people with MBIDs. The RFQ is a self-report questionnaire, which is in line with the right of autonomy, enshrined in the

Convention on the Rights of Persons with Disabilities from the United Nations in December 2006 [52], by asking the target group themselves instead of other people who would report about the target group. Moreover, with this or a further developed instrument, preliminary new knowledge about mentalising in people with MBIDs can be collected in research and practice, enabling researchers to shed new light on this relevant construct for people with intellectual disabilities and clinicians to implement this knowledge into practice. First, it becomes possible to measure the current level of mentalising so that interventions or treatment can be tailored accordingly, and more customisations can be provided. This increases the chance of an intervention being successful [11]. Finally, by being able to appropriately measure mentalising and design a successful intervention, the social functioning of this target group can be improved, which is important to the quality of life of people with MBIDs [53]. A side note for clinicians in practice is that the questionnaire is officially designed for people with BPD because impaired mentalising can be a symptom or sign of psychopathology [17]). However, the questionnaire is primarily designed to assess an individual's capacity for mentalising; therefore, it is advised to also conduct additional investigations when BPD is suspected.

## Conclusions

Overall, the results of the present study provide first insights into the psychometric properties of the adapted Dutch RFQ for assessing mentalising in adults with MBIDs. Additional other-oriented items shed a new light on the RFQ and seems valuable for people with MBIDs. The use of the RFQ in this target group is new and offers opportunities for research about mentalisation in this population that can be expanded through more research and knowledge from different (new) measures assessing mentalising.

## Supporting information

**S1 File. The Reflective Functioning Questionnaire for people with mild to borderline intellectual disabilities.**
(PDF)

## Acknowledgments

We are thankful for the participation of the respondents in the study. We thank Suze van Wijngaarden for developing and compiling the questionnaires and setting up the study, and Mirjam Wouda as the scientist practitioner from the care organisation Ons Tweede Thuis on the umbrella project 'Social relationships and ICT' for her support during data collection. We are grateful for the support of co-researchers Mark Meekel, Martijn Keesenberg and Gavin Buijtenhek for their help and feedback on the questionnaires. Finally, we are thankful for the help of master's degree students from Vrije Universiteit Amsterdam department of Clinical Child and Family studies for data collection and data coding.

## Author Contributions

**Conceptualization:** Suzanne D. M. Derks, Agnes M. Willemen, Cis Vrijmoeth.

**Data curation:** Suzanne D. M. Derks.

**Formal analysis:** Suzanne D. M. Derks, Agnes M. Willemen.

**Funding acquisition:** Paula S. Sterkenburg.

**Investigation:** Suzanne D. M. Derks, Paula S. Sterkenburg.

**Methodology:** Suzanne D. M. Derks, Agnes M. Willemen, Cis Vrijmoeth, Paula S. Sterkenburg.

**Project administration:** Suzanne D. M. Derks, Paula S. Sterkenburg.

**Supervision:** Agnes M. Willemen, Paula S. Sterkenburg.

**Validation:** Suzanne D. M. Derks, Agnes M. Willemen, Cis Vrijmoeth, Paula S. Sterkenburg.

**Visualization:** Suzanne D. M. Derks.

**Writing – original draft:** Suzanne D. M. Derks, Agnes M. Willemen, Cis Vrijmoeth, Paula S. Sterkenburg.

**Writing – review & editing:** Suzanne D. M. Derks, Agnes M. Willemen, Cis Vrijmoeth, Paula S. Sterkenburg.

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
