## [Decision Letter · Decision Letter 0]

24 Feb 2023

PONE-D-22-18695Psychometric properties of an adapted, Dutch version of the Reflective Functioning Questionnaire (RFQ) for people with mild to borderline intellectual disabilitiesPLOS ONE

Dear Dr. Derks,

Thank you for submitting your manuscript to PLOS ONE. After careful consideration, we feel that it has merit but it should be improved and maybe also modified. Therefore, we invite you to submit a revised version of the manuscript that addresses the points raised during the review process. As pointed out by Reviewer 2, the procedure and the statistical properties are less than ideal - shouldn't the title be rather something like Lessons learned from the adaptation of the Reflective Functioning Questionnaire (RFQ) for Dutch people with mild to borderline intellectual disabilities? Of course, it would shift the focus and would require some modifications. But it would give you an opportunity to share the current version and its psychometric properties while keeping the door open for an improved version. Please submit your revised manuscript by Apr 10 2023 11:59PM. If you will need more time than this to complete your revisions, please reply to this message or contact the journal office at plosone@plos.org. Please include the following items when submitting your revised manuscript:A rebuttal letter that responds to each point raised by the academic editor and reviewer(s). You should upload this letter as a separate file labeled 'Response to Reviewers'.A marked-up copy of your manuscript that highlights changes made to the original version. You should upload this as a separate file labeled 'Revised Manuscript with Track Changes'.An unmarked version of your revised paper without tracked changes. You should upload this as a separate file labeled 'Manuscript'.

We look forward to receiving your revised manuscript.

Kind regards,

Frantisek Sudzina

Academic Editor

PLOS ONE

Journal Requirements:

Reviewers' comments:

Reviewer's Responses to Questions

**Comments to the Author**

1. Is the manuscript technically sound, and do the data support the conclusions?

Reviewer #1: Yes

Reviewer #2: Partly

2. Has the statistical analysis been performed appropriately and rigorously? 

Reviewer #1: Yes

Reviewer #2: Yes

3. Have the authors made all data underlying the findings in their manuscript fully available?

Reviewer #1: No

Reviewer #2: No

4. Is the manuscript presented in an intelligible fashion and written in standard English?

Reviewer #1: Yes

Reviewer #2: Yes

5. Review Comments to the Author

Reviewer #1: The authors address an interesting and important topic: they investigate psychometric properties of the modified version of Reflective Functioning Questionnaire in the sample of people with mild to borderline intellectual disabilities. The modified by Authors version of the RFQ seems to be a promising measure for conducting further research concerning the mentalizing abilities in people with intellectual disabilities. What follows are my reactions and suggestions as I read through the manuscript:

It would be worth to provide more information about the Reflective Functioning Questionnaire in the Introduction in particular explain what were the characteristics and limitations of the original RFQ-8 and why unidimensional structure of the RFQ-8 was proposed by Müller et al. and by Woźniak-Prus et al.

Better rationale and justification should be provided for the hypothesis that non-significant correlations between RFQ-8 and other-oriented dimensions of mentalizing are expected (or Authors should reconsider this hypothesis). Several studies show that the ability to understand one's own mental states and ability to understand the mental states of other people are related to each other.

Previous studies (Müller et al, Woźniak-Prus et al.) tested if the RFQ-8 has U-shaped associations with maladaptive characteristics since very low levels of uncertainty could indicate a sense of overconfidence in understanding the mental states of other people, which could be one of the features of hypermentalizing. They did not found U-shaped characteristics, however it would be interesting to: (i) either test if the RFQ-other and RFQ-self subscale have U-shape associations with other variables e.g. with autistic traits; (ii) or to reflect on this topic in the future studies section. In particular, high scores on such items as “It is easy for me to know what other people are thinking and feeling. “ could indicate hypermentalizing.

Reviewer #2: This submission aimed to adapt the Reflective Functioning Questionnaire for use in individuals with mild to borderline intellectual disabilities (MBID) in Dutch language and to validate the new measure in a sample of 159 participants from the target population. I must admit that this review was very difficult for me because I liked the study in general and found it commendable that the authors aim to improve assessment for individuals with MBID, which can have problems with completing standard questionnaires due to too complicated items. The authors also evidently put much effort in this study. However, I unfortunately don't think that the manuscript can be published in its current state. Although my review may be disappointing to read, I strongly encourage the authors to continue with this project and ideally collect additional data that target the limitations of this study. I do believe that the field would benefit from a mentalizing measure that is suitable for individuals with MBID.

The major problem of this study is that in my view, it unfortunately cannot be deemed a proper validation study. The authors first translated into Dutch and then modified (i.e., simplification of items; adding additional items) their measure of interest - the RFQ - to make it suitable for use in the MBID population. However, the other measures used were adapted for this study as well in the same vein. The autism questionnaire was modified (simplified) for individuals with MBID here (although no reference is provided for a Dutch version, so that it was not entirely clear whether this measure was translated as well), the perspective-taking scale was translated AND modified (simplified and items removed) like the RFQ, and if I understood correctly, even the emotion recognition task was not used in a validated version here ("For this study, 50 photographs were selected", page 13). Thus, all validation criteria except for the theory of mind task were apparently used in their respective versions for the first time in this study. This leads to the circumstance that we do not know whether correlations with these measures tell us anything about the validity of the adapted RFQ, because the intended validity criteria are not validated themselves. We simply don't know whether any of these measures still reflect their intended constructs in an MBID sample. This is also evident in the results, where the adapted autism questionnaire exhibited insufficient internal consistency of alpha = .42, and the translated and adapted perspective-taking scale suffered from the same problem with alpha = .54. I'm really sorry, but I fear that this problem jeopardizes the whole study, because the validity tests are not valid. In light of this, I will not comment further on specific correlations that were reported in this manuscript.

I liked the simplifications the authors made to the original items. However, with respect to the item content, this being the RFQ item set, the question arises whether all of these items are valid indicators of mentalizing. As the authors mention in the discussion section (page 24), in Müller et al. (2022; https://doi.org/10.1080/00223891.2021.1981346), we can see that some of the RFQ items may assess what one might call consequences of impaired mentalizing rather than mentalizing in a narrow sense. For example, items starting with "When I am angry, …" seem to converge more strongly with measures of impulsivity than with other measures of mentalizing the self. This problem likely also pertains to their simplified versions used here, but there is no way of testing it in the present data. It seems noteworthy, however, that these items defined the self factor with the highest factor loadings in this study, so it would be interesting to see correlations with an impulsivity measure.

A crucial aspect for this demographic likely is the actual assessment, that is, filling out a self-report questionnaire. In the procedure section (page 9), the authors write "Independent research assistants supported the participants in completing the digital questionnaire following a standardised protocol" - what does this mean exactly? To what degree did the assistants influence the assessment procedure? Did they help with selecting response options? I think much more detail needs to be reported here, because this is a very sensitive and important information about the independence of resulting test scores.

I found it somewhat unclear which responses were used for the factor analyses of the RFQ. The authors described in the methods section (page 10) that "the answering options on the 7-point scale were split into two steps. First, participants could choose to score disagree, neutral, or agree. Second, the choices ‘disagree’ and ‘agree’ were split up into strongly, quite a bit, and somewhat". Do I understand it correctly that this two-step response process still resulted in one rating of 1-7? Frankly, I am not sure how big of an issue this is, but in principal, the response process was altered, whereas using the resulting scores in factor analysis treats these as if stemming from one response process only…

In general, it should be noted that the sample size of N = 159 for the main analyses - despite the understandable difficulty to reach sufficient sample sizes from a specific target population - is not large enough for stable estimation of correlations (Schönbrodt & Perugini, 2013; https://doi.org/10.1016/j.jrp.2013.05.009). In addition, what was the sample size rationale for the subsample for the retest?

Page 11: Müller et al. (2022; https://doi.org/10.1080/00223891.2021.1981346) found support for a unidimensional factor structure and adequate internal consistency in a German inpatient sample, a German young adult sample, and a US sample (omega = .87). The unidimensional structure was also confirmed in a representative German sample in Spitzer et al. (2021, omega = .82 for 8 items; https://doi.org/10.1055/a-1234-6317). The authors might want to add this information.

The finding that the factor structure becomes two-dimensional and reflects self- and other-mentalizing nicely aligns with findings for the CAMSQ (Müller et al., 2021; https://doi.org/10.1177/10731911211061280) which also consists of these two factors. Thus, once balanced item content is used (as compared to the original RFQ that mostly pertains to self-mentalizing), self and other appear to emerge as the two major dimensions of mentalizing.

6. PLOS authors have the option to publish the peer review history of their article (what does this mean?). If published, this will include your full peer review and any attached files.

Reviewer #1: No

Reviewer #2: **Yes: **Sascha Müller

---

## [Author Response · Author response to Decision Letter 0]

30 May 2023

Response to the reviewers.

We thank the reviewers for reviewing our manuscript ‘Psychometric properties of an adapted, Dutch version of the Reflective Functioning Questionnaire (RFQ) for people with mild to borderline intellectual disabilities’. We very much appreciate the feedback we received. We are also grateful for the opportunity to revise our manuscript. We believe that the manuscript has been improved further.

Please receive our revision and the changes to each comment below. In response to the editors’ suggestions, we adjusted the title of our manuscript to ‘Lessons learned from the adaptation of the Reflective Functioning Questionnaire (RFQ) for Dutch people with mild to borderline intellectual disabilities’.

All comments have been integrated in the new version, with the changes highlighted with yellow. The manuscript was thoroughly checked on grammatical and typing errors which we did not highlight for the reviewers’ convenience. 

Yours sincerely,

The Authors

Reviewer 1

We greatly thank reviewer 1 for the compliment, suggestions, and comments.

1.1) Comment: It would be worth to provide more information about the Reflective Functioning Questionnaire in the Introduction in particular explain what were the characteristics and limitations of the original RFQ-8 and why unidimensional structure of the RFQ-8 was proposed by Müller et al. and by Woźniak-Prus et al.

The Introduction now contains more information about the Reflective Functioning Questionnaire: ‘The RFQ consist of 8 items divided across two subscales: ‘Certainty about mental states’ and ‘Uncertainty about mental states’. The Certainty subscale assesses the development of complex models of the mind that are inconsistent with observable evidence (i.e., hypermentalising). The Uncertainty subscale assesses the great difficulty with developing complex models of the mind of the self and/or others (i.e., hypomentalising). Genuine mentalising is the optimal level of mentalising in between hyper- and hypomentalising, characterised by a balanced stance of knowing and not always knowing the mental states of themselves and others [18]. Four of the eight items were used in both subscales. Therefore, these items were double scored in opposite direction. However, this double scoring resulting in a two-factor structure is psychometrically questionable because items in a factor analysis are assumed to be independent. These challenges were noted in the studies of Müller et al. [19], Spitzer et al. [20] and Woźniak-Prus et al. [21]. Therefore, the authors took a step back and conducted an exploratory factor analysis to assess factor structure of the RFQ. The authors showed that the eight items fit well as a unidimensional construct [19–21].’ (Introduction, p. 5)

1.2) Comment: Better rationale and justification should be provided for the hypothesis that non-significant correlations between RFQ-8 and other-oriented dimensions of mentalizing are expected (or Authors should reconsider this hypothesis). Several studies show that the ability to understand one's own mental states and ability to understand the mental states of other people are related to each other.

For correlations between the RFQ and the other-oriented dimensions, mixed results are found in previous studies. For example, correlations between the RFQ and a Theory of Mind test were assessed in three studies, finding either (1) weak, non-significant correlations (Morandotti et al., 2018 ), or (2) weak, significant correlations in a non-clinical sample and weak, non-significant correlations in a clinical sample (Fonagy et a., 2016 ) and (3) weak, significant correlations with Certainty but weak, non-significant correlations with Uncertainty (Cucchi et al., 2018). For perspective taking, non-significant (Fonagy et al., 2016) and significant correlations (Cucchi et al., 2018; Fonagy et al., 2016) were found. In line with doing a more explorative study, we reformulated the hypothesis: ‘Third, correlations are examined, comparing the RFQ-8 and the extended RFQ with autistic traits. Stronger correlations were expected for the extended RFQ and autistic traits, because the other-oriented focus is also represented in autistic traits (e.g., greater challenge with other people's thoughts and feelings). In addition, associations between the RFQ-8 and the extended RFQ and perspective taking, emotion recognition and Theory of Mind were investigated. Weaker correlations with the RFQ-8 compared to the extended RFQ are expected as the RFQ-8 is primarily internal and self-oriented while the other concepts reflect the external, other-oriented (e.g., emotion recognition and Theory of Mind) or internal, other-oriented (e.g., perspective taking) dimensions of mentalising.’ (Introduction, p. 6)

1.3) Previous studies (Müller et al, Woźniak-Prus et al.) tested if the RFQ-8 has U-shaped associations with maladaptive characteristics since very low levels of uncertainty could indicate a sense of overconfidence in understanding the mental states of other people, which could be one of the features of hypermentalizing. They did not found U-shaped characteristics, however it would be interesting to: (i) either test if the RFQ-other and RFQ-self subscale have U-shape associations with other variables e.g. with autistic traits; (ii) or to reflect on this topic in the future studies section. In particular, high scores on such items as “It is easy for me to know what other people are thinking and feeling. “ could indicate hypermentalizing.

Thank you for the interesting suggestion. We agree with the reviewer that very low levels of uncertainty could indicate hypermentalising. With a U-shaped association between the RFQ subscales and the maladaptive characteristics of autistic traits, it is suggested that high scores on autistic traits are associated with both hypo- and hypermentalising. Indeed, previous studies showed that typically developing persons with autism spectrum disorder tend to show both hypo- and hypermentalising (e.g., Isaksson et al., 2019). A first look at a scatterplot with the RFQ Other and autistic traits and a scatterplot with the RFQ Self and autistic traits revealed no U-shaped associations between both variables. Moreover, in our study, the AQ-10 for people with MBIDs, assessing autistic traits, was adapted, and used for the first time. A low Cronbach’s alpha was revealed, questioning the reliability of the questionnaire. Therefore, we would suggest this a topic for future studies, with a valid instrument for autistic traits. We added to the discussion: ‘In addition, Müller et al. [19], Spitzer et al. [20] and Woźniak-Prus et al. [21] also examined U-shaped associations between the RFQ and maladaptive characteristics such as anxiety, signs of psychopathology and mental health problems, since very low scores on the RFQ may indicate hypermentalising (e.g., being overconfident in understanding mental states). Future studies are invited to perform analyses to further investigate the possibility of a U-shaped association for people with intellectual disabilities with a well-validated measure of autistic traits.’ (Discussion, p. 25)

Reviewer 2

We thank Reviewer 2 for the compliment on our work. We also thank the reviewer for sharing the point of view to our study and thank for the helping, clear comment. We will respond to each comment below. 

2.1) The major problem of this study is that in my view, it unfortunately cannot be deemed a proper validation study. The authors first translated into Dutch and then modified (i.e., simplification of items; adding additional items) their measure of interest - the RFQ - to make it suitable for use in the MBID population. However, the other measures used were adapted for this study as well in the same vein. The autism questionnaire was modified (simplified) for individuals with MBID here (although no reference is provided for a Dutch version, so that it was not entirely clear whether this measure was translated as well), the perspective-taking scale was translated AND modified (simplified and items removed) like the RFQ, and if I understood correctly, even the emotion recognition task was not used in a validated version here ("For this study, 50 photographs were selected", page 13). Thus, all validation criteria except for the theory of mind task were apparently used in their respective versions for the first time in this study. This leads to the circumstance that we do not know whether correlations with these measures tell us anything about the validity of the adapted RFQ, because the intended validity criteria are not validated themselves. We simply don't know whether any of these measures still reflect their intended constructs in an MBID sample. This is also evident in the results, where the adapted autism questionnaire exhibited insufficient internal consistency of alpha = .42, and the translated and adapted perspective-taking scale suffered from the same problem with alpha = .54. I'm really sorry, but I fear that this problem jeopardizes the whole study, because the validity tests are not valid. In light of this, I will not comment further on specific correlations that were reported in this manuscript.

We agree with the reviewer that, for a proper validation study, other measures are desirable to be well validated and not adapted to be used as validity criteria or ‘golden standards’. With some measures adapted, the correlational analysis has limitations. However, we faced the challenge that for people with MBIDs, questionnaires are not yet adapted or developed, especially on this topic. Therefore, we were urged to make the necessary adaptations. 

- Simplify the AQ-10 for individuals with MBIDs. Reference of the validation study of the Dutch version of the AQ was added (p. 12). 

- We apologise for the mistake made but the perspective-taking scale was only simplified (text is adjusted). Results on internal consistency urged us to take a closer look and remove two items. 

- The RaFD is indeed not yet validated for people with MBIDs. However, the selection of the photo’s was already done in the study of Van Rest. We clarified this in the manuscript. 

As these questionnaires were assessed for the first time in people with MBIDs, results also could (and did) reveal challenges with the questionnaires (i.e., low internal consistency). We agree with the reviewer and are happy with the suggestion of the editor to approach the study as an ‘first, explorative, lessons learned study’. Therefore, we modified our manuscript accordingly. For example, we are careful in using the terms ‘reliability’ and ‘validity’, and added to the Discussion: ‘Second, given the novelty of the study, the correlational analyses are restricted by unmodified (AQ-10, PT subscale) and unvalidated (AQ-10, PT subscale, RaFD, Frith Happé animations test) instruments for people with MBIDs. As these instruments were assessed for the first time in this target group, results also could and did reveal challenges with the instruments (i.e., low internal consistency). This has limited statements about the psychometric properties of the RFQ.’ (Discussion, p. 24) 

And 

‘In sum, this study shows that reflective functioning measured with the RFQ can provide complementary insights about mentalising of a certain person when added to a battery of other measures assessing important skills for mentalising. However, following Müller et al. [19], Spitzer et al. [20] and Woźniak-Prus et al. [21], the RFQ alone does not seem to be sufficient to assess (the lack of) complex mentalising, also for people with MBIDs.’ (Discussion, p. 23)

2.2) I liked the simplifications the authors made to the original items. However, with respect to the item content, this being the RFQ item set, the question arises whether all of these items are valid indicators of mentalizing. As the authors mention in the discussion section (page 24), in Müller et al. (2022; https://doi.org/10.1080/00223891.2021.1981346), we can see that some of the RFQ items may assess what one might call consequences of impaired mentalizing rather than mentalizing in a narrow sense. For example, items starting with "When I am angry, …" seem to converge more strongly with measures of impulsivity than with other measures of mentalizing the self. This problem likely also pertains to their simplified versions used here, but there is no way of testing it in the present data. It seems noteworthy, however, that these items defined the self factor with the highest factor loadings in this study, so it would be interesting to see correlations with an impulsivity measure.

Unfortunately, we didn’t include an impulsivity measure in the questionnaire. The reason for this was our assumption that the burden of completing questionnaires and the attention span of our target group would limit the number of questionnaires that could be included. In our discussion section, we invite future research to elaborate on this: ‘Moreover, as recently discussed by Müller et al. [19], some items of the RFQ may overlap with characteristics such as impulsivity and emotional lability, questioning the accuracy of the RFQ to measure mentalising. With this in mind, Müller et al. [19] developed the CAMSQ [42]. Future studies are invited to first replicate this study and investigate the convergence of the RFQ with impulsivity, and second to examine the applicability of new questionnaires as the CAMSQ [42] for people with MBIDs.’ (Discussion, p. 24). 

2.3) A crucial aspect for this demographic likely is the actual assessment, that is, filling out a self-report questionnaire. In the procedure section (page 9), the authors write "Independent research assistants supported the participants in completing the digital questionnaire following a standardised protocol" - what does this mean exactly? To what degree did the assistants influence the assessment procedure? Did they help with selecting response options? I think much more detail needs to be reported here, because this is a very sensitive and important information about the independence of resulting test scores.

We agree with the reviewer that with the support of independent research assistants, the independence of resulting test scores need to be more clarified. Specifically, that (1) providing sensitive answers can cause undue under pressure and (2) personal opinions of a research assistant could influence the (self-reporting) answers of the participants. To maximise the independence of the resulting test scores, (1) we advised participants that that their answers would not be shared with other people and (2) stimulated thinking through extra explanations and by that there were no wrong answers. We clarified this in our Method section: ‘Independent research assistants supported the participants in completing the digital questionnaire following a standardised protocol. According to this protocol, research assistants were instructed to help the participants with the digital questionnaires provided by Qualtrics software (i.e., digital support) to stimulate thinking through extra explanations (see Supporting Information). Moreover, they appointed that it is about their thoughts and feelings and that there were no wrong answers. In addition, participants are ensured that their answers would not be shared with other people.’ (Method, p. 8, 9).

2.4) I found it somewhat unclear which responses were used for the factor analyses of the RFQ. The authors described in the methods section (page 10) that "the answering options on the 7-point scale were split into two steps. First, participants could choose to score disagree, neutral, or agree. Second, the choices ‘disagree’ and ‘agree’ were split up into strongly, quite a bit, and somewhat". Do I understand it correctly that this two-step response process still resulted in one rating of 1-7? Frankly, I am not sure how big of an issue this is, but in principal, the response process was altered, whereas using the resulting scores in factor analysis treats these as if stemming from one response process only…

Indeed, the two-step response process still resulted in one rating of 1-7. In the first step, the scores ‘disagree’ and ‘agree’ are used as ‘click through’ answers categories. We agree with the reviewer that in principle the response process was altered. is. We consciously decided to split the response process, based on advice provided by Finlay and Lyons for people with intellectual disabilities (i.e., for multiple-choice format questions break the question into two stages) to help the participants answer the question. Moreover, we argue that the resulting score stems from one response process. That is, the resulting score for ‘disagree’ or ‘agree’ was only a refinement of the corresponding answer category. In other words, the scores after ‘disagree’ or ‘agree’ could not result in completely different scores (e.g., first selecting ‘disagree’ could not result in ‘agree’ in the second response option). 

2.5) In general, it should be noted that the sample size of N = 159 for the main analyses - despite the understandable difficulty to reach sufficient sample sizes from a specific target population - is not large enough for stable estimation of correlations (Schönbrodt & Perugini, 2013; https://doi.org/10.1016/j.jrp.2013.05.009). In addition, what was the sample size rationale for the subsample for the retest?

We added to the Discussion: ‘First, considering a small effect size (.10), the sample size of the study (N = 159) is not sufficient to establish significant correlations (to .40) [47]. Additional data of adults with MBIDs is necessary to strengthen the correlations.’ (Discussion, p. 23). 

The rationale for the sample size of the subsample for the test-retest was based on the web-based sample size calculator. With a minimum acceptable reliability of .50, expected reliability of .70, α = .05 two-tailed, β = .80, k = 2 and expected dropout rate of 10%, the sample size calculated was 88 to recruit, 79 with dropout. We added to the Method section: ‘Test-retest sample size (n = 83) was satisfactory to detect a minimum acceptable ICC of .50, with an expected ICC of .70, α = .05 two-tailed, β = .80, k = 2 and expected dropout rate of 10% [25].’ (Method, p. 7).

2.6) Page 11: Müller et al. (2022; https://doi.org/10.1080/00223891.2021.1981346) found support for a unidimensional factor structure and adequate internal consistency in a German inpatient sample, a German young adult sample, and a US sample (omega = .87). The unidimensional structure was also confirmed in a representative German sample in Spitzer et al. (2021, omega = .82 for 8 items; https://doi.org/10.1055/a-1234-6317). The authors might want to add this information.

We thank the reviewer for the suggestion provided. We added the information to our Method section: ‘Recent studies have indicated adequate psychometric properties of the original RFQ-8 as unidimensional scale in a German inpatients sample, a German young adult sample, a United States adult sample [19] and a German adult sample [20], with McDonald’s omegas of .79, .82, .87 and .82 respectively. Additionally, a Cronbach’s alpha of .75 was found in a Polish sample of students [21].’ (Method, p. 11).

2.7) The finding that the factor structure becomes two-dimensional and reflects self- and other-mentalizing nicely aligns with findings for the CAMSQ (Müller et al., 2021; https://doi.org/10.1177/10731911211061280) which also consists of these two factors. Thus, once balanced item content is used (as compared to the original RFQ that mostly pertains to self-mentalizing), self and other appear to emerge as the two major dimensions of mentalizing.

We thank the reviewer for the comment and are delighted to notice that these results can complement each other. We are also interested in the CAMSQ, which was not yet available when our study was conducted (i.e., data collection in 2018-2019). We enriched our Discussion with: ‘The two-factor structure with the subscales ‘RFQ Self’ and ‘RFQ Other’ is in line with recent findings of Müller et al. [19], who developed a new mentalising measure called the Certainty About Mental States Questionnaire (CAMSQ) [42]. Both results underline the importance of including the dimensions of ‘the self’ and ‘the other’ when assessing mentalising. Creating a specific RFQ Other subscale for people with MBIDs seems moreover valuable as they tend to have more difficulties reflecting on the feelings and thoughts of others and taking the perspective of others [43, 44].’ (Discussion, p. 21).

Again, we thank the reviewers for their time and efforts to improve the paper. We hope that the contents of this paper inspire further research in this field.

---

## [Editor Report · Decision Letter 1]

13 Jun 2023

Lessons learned from the adaptation of the Reflective Functioning Questionnaire (RFQ) for Dutch people with mild to borderline intellectual disabilities

PONE-D-22-18695R1

Dear Dr. Derks,

We’re pleased to inform you that your manuscript has been judged scientifically suitable for publication and will be formally accepted for publication once it meets all outstanding technical requirements.

Kind regards,

Frantisek Sudzina

Academic Editor

PLOS ONE
---

## [Editor Report · Acceptance letter]

19 Jun 2023

PONE-D-22-18695R1 

Lessons learned from the adaptation of the Reflective Functioning Questionnaire (RFQ) for Dutch people with mild to borderline intellectual disabilities 

Dear Dr. Derks:

I'm pleased to inform you that your manuscript has been deemed suitable for publication in PLOS ONE. Congratulations! Your manuscript is now with our production department. 

Kind regards, 

on behalf of

Dr. Frantisek Sudzina 

Academic Editor

PLOS ONE